# Predictive Significance of the ABC Score for Early Re-Hemorrhage and In-Hospital Mortality in High-Risk Variceal Bleeding among Cirrhotic Patients

**DOI:** 10.3390/diagnostics13233570

**Published:** 2023-11-29

**Authors:** Thai Doan Ky, Nguyen Thi Huyen Trang, Mai Thanh Binh

**Affiliations:** Department of Gastroenterology and Hepatology, 108 Military Central Hospital, Hanoi 10000, Vietnam; kythaitrung@gmail.com (T.D.K.); drtrang1009@gmail.com (N.T.H.T.)

**Keywords:** ABC score, early re-hemorrhage, in-hospital mortality

## Abstract

(1) Background: Upper gastrointestinal bleeding due to ruptured varices is a severe complication in patients with cirrhosis, with high rates of recurrent hemorrhage and in-hospital mortality. This study aimed to evaluate the value of the ABC score in predicting two events among 201 cirrhotic patients with high-risk variceal hemorrhage. (2) Methods: The ABC score was calculated and categorized into risk groups of patients, and the association between the ABC score and the rates of early hemorrhagic recurrence and clinic mortality were analyzed. (3) Results: Among 201 patients, 8.0% experienced early rebleeding within five days of admission, and 10.4% died in the hospital. Patients who experienced events had higher average ABC scores compared to those who did not experience these events (*p* < 0.001), especially in the high-risk group (with ABC score ≥ 8). The ABC score showed an excellent predictive value for in-hospital mortality with an AUROC of 0.804, with the optimal cutoff point being 8 points. Additionally, the ABC score demonstrated an acceptable predictive value for early rebleeding with an AUROC of 0.744, and the best cutoff point was 9 points. (4) Conclusions: The ABC score is closely associated with the rates of early re-hemorrhage and in-hospital mortality in cirrhotic patients with variceal bleeding. This scoring system has the potential for clinical application, aiding in early risk stratification for recurrent bleeding and mortality and allowing for more aggressive interventions in high-risk cases.

## 1. Introduction

Upper gastrointestinal bleeding (UGIB) is a typical medical emergency. In the United States, there are approximately 350,000 cases of UGIB admissions each year, with an incidence of about 100 per 100,000 population. There are many causes of GIB, among which hemorrhage from esophageal varices and gastric varices in portal hypertension syndrome is a frequent cause, accounting for about 30% [1,2]. Despite significant advancements in diagnosis and treatment, secondary bleeding events and mortality within the first month remain high in patients with UGIB due to ruptured esophageal varices and gastric varices at 25.7% and 15.2%, respectively. The primary reason is the failure to effectively control bleeding in the early days [3,4]. The risk assessment score is used in these patients to predict important clinical outcomes, including the need for interventions, recurrent bleeding, and mortality. These scores assist clinicians in determining the urgency of endoscopy, the level of care required, ongoing treatment, monitoring, and post-intervention prognosis. A sufficiently robust scoring system or tool for early rebleeding prognosis will bring about greater effectiveness and improved survival benefits for patients with upper gastrointestinal bleeding.

Prognosticating UGIB due to ruptured esophageal varices (ESOVs) or gastric varices (GASVs) is often challenging due to the severity of the hemorrhage and the underlying liver fibrosis. Several factors have been studied for prognostication, such as the Child–Pugh score, Model for End-Stage Liver Disease (MELD), AIMS65, and Rockall score. However, these factors have limitations, such as subjective elements and complex calculations, and some reports indicate that their predictive value for rebleeding and mortality risk is not yet sufficiently high [5].

In 2020, Laursen S.B. and colleagues conducted a multicenter international study and developed a new prognostic scoring system for UGIB called the ABC score [6]. This scoring system is based on three criteria: age, blood test results, and comorbidity. The score ranges from 0 to 18 points, categorizing the risk into low (≤3 points), moderate (4–7 points), and high (≥8 points) levels. The 30-day mortality rates for high-risk UGIB patients in these three risk groups were 1%, 7%, and 25%, respectively [6]. Additionally, compared to other scoring systems like AIMS65, Rockall, and Glasgow-Blatchford, the ABC score demonstrated similar or significantly better predictive ability in predicting 30-day mortality and early recurrent hemorrhage rates in patients with gastrointestinal bleeding [6,7,8]. Following the study by Laursen S.B., several other studies have also reinforced the solid predictive value of the ABC score in predicting 30-day mortality in patients with upper gastrointestinal bleeding with an accurately predicted AUROC of approximately 0.85 [7,9,10]. However, there is currently a limitation of data available on the value of the ABC score in predicting healthcare facility mortality (typically within about a week) and early re-hemorrhage rates (within five days).

In Vietnam, estimates indicate an incidence of about 100 cases per 100,000 individuals with gastrointestinal bleeding, including upper and lower hemorrhage, accompanied by a mortality rate that falls within the range of 10% to 20% (derived from insights provided by domestic scientific publications). Few tools are available to predict the mortality rate among patients with lower gastrointestinal bleeding [11]. Similarly, a few scoring systems are believed to be valuable in predicting the mortality rate among patients with upper gastrointestinal bleeding [12]. In contrast, many UGIB patients visit hospitals each year with high mortality. On the other hand, no study has applied the ABC score to the prognosis of patients with liver cirrhosis complicated by UGIB due to the rupture of esophageal or gastric varices. Therefore, our study aims to evaluate the value of the ABC score in predicting immediate re-hemorrhage and in-hospital mortality in patients with liver cirrhosis complicated by UGIB due to esophageal and/or gastric varices.

## 2. Materials and Methods

### 2.1. Patients

This study included 201 patients diagnosed with upper gastrointestinal bleeding (UGIB) due to ruptured esophageal varices and gastric varies on a background of liver fibrosis. The patients were treated at the Institute of Gastroenterology, 108 Military Central Hospital in Vietnam, from March 2022 to May 2023.

Inclusion criteria: Patients admitted to the hospital with symptoms of hematemesis and/or melena underwent clinical and laboratory examinations and an upper gastrointestinal endoscopy and were diagnosed with UGIB due to ruptured esophageal (ESOVs) and/or gastric varices (GASVs) on a background of cirrhosis. The participants were diagnosed with liver fibrosis, particularly alcohol-related, based on the national guidelines. These patients had a history of chronic liver damage (long-term alcohol use), impaired liver function, portal hypertension syndrome (indicated by esophageal varices and/or variceal dilation), and diagnostic imaging confirming liver fibrosis on ultrasound/FibroScan/CT scanner [13]. ESOV or GASV bleeding was confirmed by upper GI endoscopy with the following findings: Presence of ESOVs or GASVs that are oozing or spurting. Direct visualization of blood arising from ESOVs or GASVs (spurting or oozing). Presence of ESGVs or GASVs with signs of recent bleed (stigmata) such as white nipple sign or overlying clot. Presence of ESGVs or GASVs with red signs plus overt blood in the stomach in the absence of another source of bleeding.

Exclusion criteria: UGIB caused by other factors such as Mallory–Weiss syndrome and esophageal or gastrointestinal ulcers; UGIB due to ruptured veins in conditions other than increased portal pressure like portal hypertension syndrome unrelated to cirrhosis; and patients who did not consent to participate in this study. Patients used anticoagulants.

### 2.2. Methods

#### 2.2.1. Research Design: A Retrospective Study

All patients were treated according to a standardized protocol based on their pathological condition, including volume resuscitation, hemostasis measures (bed rest, endoscopic intervention, hemostatic drugs, portal-pressure-reducing agents), indicated blood transfusions, prevention of hepatic encephalopathy, and infection following the guidelines of the Baveno VII Consensus (2022) [14]. All patients underwent endoscopy and hemostatic intervention following the national guidelines: esophageal variceal bleeding was managed by rubber band ligation, and esophageal variceal bleeding was treated with injection of varices using Histoacryl [13]. The patients who failed with endoscopic hemostatic intervention underwent emergency TIPS (Transjugular Intrahepatic Portosystemic Shunt) [14]. The monitoring of subsequent hemorrhage and in-hospital mortality was conducted during the treatment period at the hospital. Recurrent hemorrhage was defined as the occurrence of hematemesis after 2 h or more post-treatment with medication or endoscopic hemostasis or a decrease in Hemoglobin levels by 30 g/L or more (approximately 9% Hematocrit) within 24 h in patients who did not receive blood transfusion [15]. Early rebleeding was defined as the occurrence of hematemesis episodes within the first five days [14,16].

Calculation of the ABC score: proposed by author Laursen S.B. and colleagues (2020), it includes the factors of age, blood test, and comorbidity with values ranging from 0 to 18 points [6]. The score ranges from 0 to 18 points, categorizing the risk into low (≤3 points), moderate (4–7 points), and high (≥8 points) levels.

#### 2.2.2. Statistical Analysis

Data processing and analysis were conducted using SPSS 25.0 medical statistics software, MedCalc software 22.016, and GraphPad Prism 9.0. Proportions were compared using the Chi-square test or Fisher’s Exact Test. Receiver operating characteristic (ROC) curves were constructed, and the area under the curve (AUC) was determined to identify a suitable cutoff point with corresponding sensitivity and specificity (the cutoff point is the point at which the J value is maximized, with J = sensitivity + specificity − 1). Using the determined cutoff point, a 2 × 2 table was used to reassess sensitivity (Se), specificity (Sp), positive predictive value (PPV), and negative predictive value (NPV). Results were considered statistically significant at *p* < 0.05.

## 3. Results

### 3.1. Baseline Characteristics of the Patients with Upper Gastrointestinal Bleeding

The demographic and endoscopic characteristics of the 201 patients with upper gastrointestinal bleeding are described in Table 1.

The average age of the patient group in this study was 56.4 ± 10.8, with males comprising the majority (92.0%), with a male-to-female ratio of 11.6/1. Among them, 115 (57.2%) had alcoholic liver cirrhosis, followed by hepatitis B-related cirrhosis at 18.4%. Additionally, 152 patients (75.6%) had MELD scores ≤ 15. Liver function, as classified by Child–Pugh scores, was distributed as 21.4%, 49.8%, and 28.8% for classes A, B, and C, respectively. Moreover, 57.2% of those had experienced previous episodes of variceal hemorrhage caused by esophageal and/or gastric varices rupture. Common admission symptoms were hematemesis and melena (51.2%).

The results of upper gastrointestinal endoscopy were as follows: Grade III esophageal varices were observed in 80.6% of patients, and gastric varices were observed in 60.7% of patients. Among them, 41 patients (20.4%) were identified to have actively bleeding varices structures requiring emergency hemostatic intervention.

In the outcome of the observation, in Table 2, out of the total 201 patients in this study, 16 patients (8.0%) experienced early re-hemorrhage within five days. Particularly noteworthy is the in-hospital mortality rate, despite the application of all blood-stopping measures and blood transfusions, which reached a high of 10.4%. The primary cause of death (16/21) was severe hemorrhagic shock, multi-organ failure, and profound metabolic derangements, despite successful hemostatic intervention.

### 3.2. The Correlation between ABC Score and the Risk of Occurrences of Early Rebleeding and In-Hospital Mortality

The variables of the ABC score are shown in Table 3. The mean ABC score of the study group was 6.3 ± 2.5, predominantly concentrated in the moderate- and high-risk categories at 62.2% and 26.9%, respectively.

The relationship between the ABC score and risk of enabled re-hemorrhage and in-hospital mortality is shown in Table 4. The group of patients with early recurrent bleeding or in-hospital mortality had significantly higher average ABC scores than those without corresponding outcomes (9.1 vs. 6.1 and 9.1 vs. 6.0, respectively, *p* < 0.001).

Patients with ABC scores ≥ 8 had recurrent bleeding rates of 20.4% and healthcare facility mortality rates of 31.5%. Patients with an increasing risk of early re-hemorrhage and in-hospital fatality showed statistically significant trends according to the ABC scores (*p* < 0.05).

### 3.3. The Value of the ABC Score in Predicting Early Rebleeding and In-Hospital Mortality in UGIB Patients

The predictive value of the ABC scores for the risk of re-hemorrhage and in-hospital mortality is summarized in Table 5 and Figure 1.

The ABC score has a good value in predicting institutional death with an area under the receiver operating characteristic curve (AUROC) of 0.804; 95%CI: 0.70–0.91; *p* < 0.001. At the cutoff point of ABC score = 8 points, it has the best predictive value for hospital-based mortality, with a sensitivity of 81% and specificity of 79.4%.

Additionally, this scoring system holds an acceptable predictive value for immediate hemorrhage within five days (with an AUROC of 0.744; 95%CI: 0.59–0.89; *p* = 0.002). The cutoff point of ABC score = 9 points is the best threshold for predicting early recurrent bleeding, with a sensitivity of 62.5% and specificity of 84.9%.

## 4. Discussion

The average age of the patients in this study was 56.4 ± 10.8 years, with males constituting the majority (93.4%) and a male-to-female ratio of 14.5:1. Among the patients, 87.6% had a history of liver cirrhosis, and 57.2% had previously experienced upper gastrointestinal bleeding (UGIB) due to rupture of esophageal varies (ESOVs) and gastric varies (GASVs). Our study’s results align with previous research findings, showing that liver cirrhosis is a common chronic condition in middle-aged individuals, which is more prevalent in males and often requires multiple hospitalizations due to complications, particularly UGIB resulting from ruptured varices [17,18].

The common reason for hospital admission was often a combination of hematemesis and melena (51.2%), while 23.4% experienced only hematemesis and 25.4% solely presented with melena. A study by Aluizio C.L. and colleagues in 2021 also reported symptoms of hematemesis in 78.4% of patients and black stools in 66.2% [5].

During the upper gastrointestinal endoscopy, we observed esophageal varices in 96.0% of patients, with the majority having Grade III esophageal varices (80.6%) and only 2.0% having Grade I varices. Gastric varices were detected in 60.7% of patients. Additionally, we identified 41 patients (20.4%) with active bleeding from varices requiring emergency hemostasis. Our study findings are consistent with the report by Elsafty R.E. and colleagues in 2021, where 17% of 250 patients undergoing endoscopy had active hemorrhage from varices necessitating emergency hemostasis [19]. Similarly, Robertson M. and colleagues in 2020 also identified 58 cases (26%) with active hemorrhage from varices [18].

All our patients were treated according to a standardized protocol guided by the consensus recommendations of Baveno VII (2022) [14] and the clinical national guideline [13]; however, we observed that the rates of early recurrent hemorrhage and in-hospital mortality in Vietnam remain relatively high. During the follow-up period, we noted that 8.0% of patients experienced active hemorrhage soon within 5 days of admission, and 10.4% of patients died during hospitalization. Our results are consistent with previous reports, estimating the rate of untimely recurrent bleeding to be around 10% and the rate of in-hospital mortality within 7 days to be in the range of 10–20% [20,21,22,23]. The main causes of death in our patients (16/21) were severe hemorrhagic shock, profound metabolic derangements, and multi-organ failure although they were applied much intensive hemostatic intervention. The remaining patients died due to various other reasons, such as infections, hepatic encephalopathy, and acute coronary syndrome. These are severe complications of upper gastrointestinal bleeding in patients, especially those with liver cirrhosis, who often have coagulation disorders accompanying the bleeding. Therefore, upper gastrointestinal hemorrhage due to variceal rupture remains a severe complication in patients with cirrhosis. Despite significant advancements in treatment, the mortality rate remains high primarily due to challenges in effectively controlling bleeding episodes. Consequently, early risk stratification is crucial for high-risk groups to ensure more aggressive therapeutic interventions.

The ABC score is closely associated with the occurrence of events of premature re-hemorrhage and mortality. Our study’s average ABC score was 6.3 ± 2.5, mainly concentrated in the moderate- and high-risk groups, accounting for 62.2% and 26.9%, respectively. The average ABC score in our study is higher than that in many reports by other authors. For example, the average score in the study by Li Y. et al. (2022) was 4.0 [24], Saade M.C. et al. reported 5.26 [9], and Jimenez-Rosales R. et al. (2023) reported 4.5 [25]. It is evident that the average ABC score in our study is significantly higher, and this difference may arise from the criteria for selecting patients to participate in this study. While we specifically selected patients with cirrhosis who had UGIB due to variceal rupture, other studies included patients with upper gastrointestinal bleeding due to various causes. Nevertheless, the distribution of risk factor groups according to the ABC score in our results is consistent with previous reports, primarily consisting of patients with ABC scores ≥ 4 (moderate and high risk) [9,24,25]. Furthermore, among patients experiencing either reactive hemorrhage or early mortality, the average ABC score was significantly higher than in those without these outcomes (9.1 vs. 6.1 and 9.1 vs. 6.0, *p* < 0.001, respectively).

Cirrhosis has unique clinical and laboratory features. Therefore, predicting high-risk patients is not as easy as it seems. In the ABC scoring system, liver cirrhosis is assigned 2 points. Furthermore, a majority of cirrhotic patients are evaluated as ASA 3 or ASA 4 class. Hypoalbuminemia is also common among cirrhotic subjects. In the severe forms of cirrhosis, the development of HRS or ACLF is associated with an increase in creatinine levels. Additionally, substantial gastrointestinal bleeding from varices correlates with a significant rise in blood urea nitrogen (BUN) levels. All these factors categorize cirrhotic patients into moderate- to high-risk groups according to the ABC scoring system. Therefore, the utility of the ABC score for predicting mortality among this special patient group may be subject to doubt. However, not all cirrhotic patients fit this profile, and it depends on the stage, severity, and progression of the disease. Cirrhotic patients may have higher ABC scores than those with gastrointestinal bleeding due to other causes. Nevertheless, the ABC score retains its value in predicting early rebleeding and early mortality in this patient group, with a corresponding prediction cutoff point for each outcome.

The ABC score can be applied for prognosticating untimorous re-hemorrhage and in-hospital mortality (Table 5, Figure 1). For predicting the early secondary bleeding episode, the ABC score demonstrates an acceptable prognostic value with an area under the curve (AUROC) of 0.74, and the optimal cutoff value is 9 points. On the other hand, the ABC score exhibits good prognostic performance for the prognosis of medical center death with an AUROC of 0.8, and the best cutoff value is 8 points. Our findings are also consistent with other reports worldwide. For instance, Li Y. and colleagues (2022) reported AUROC values of 0.833 for predicting mortality and 0.718 for predicting recurrent hemorrhage [24]. Jimenez-Rosales R. and colleagues (2023) observed an AUROC of 0.80 for healthcare facility mortality prediction, while the forecast for re-hemorrhage was moderate at 0.61 [25]. Similarly, Mules T.C. and colleagues (2021) applied the ABC score to 229 high-risk upper gastrointestinal hemorrhage patients in New Zealand and found that the 30-day mortality increased with escalating risk: low risk was 1.6%, medium risk was 7.5%, and high risk was 42%; the AUROC for predicting mortality was 0.85, which was the highest among various other high-risk upper gastrointestinal bleeding prognostic scores such as PNED (0.8), complete Rockall (0.75), GBS (0.71), and AIMS65 (0.70) [7].

In Vietnam, currently there are no published statistical figures on the incidence of upper gastrointestinal bleeding and its mortality rate. However, this number is estimated to be around 100 cases per 100,000 population, with a mortality rate of about 10–20% (data from domestic scientific publications). A few studied scoring systems have shown good predictive value for mortality in Vietnamese patients with gastrointestinal bleeding, such as the Oakland and Glasgow-Blatchford scores [11,12]. However, these studies have primarily been conducted on patients with lower gastrointestinal bleeding and acute nonvariceal upper gastrointestinal bleeding, with an area under the diagnostic curve of approximately 0.71 [11,12]. Therefore, based on the results of our study, the ABC score with an AUROC of 0.8 for predicting mortality and an AUROC of 0.74 for predicting early rebleeding in patients with UGIB due to ruptured esophageal varices and gastric varices demonstrates promising potential for real-world clinical application.

This study also has a few limitations: First, the sample size is minimal, and it would be beneficial to conduct further research with larger sample sizes. More extensive studies would provide more robust and comprehensive insights into the ABC score’s value in predicting secondary hemorrhage episodes and in-hospital mortality in patients with cirrhosis and high-risk upper gastrointestinal bleeding. Second, no analysis has compared the value of the ABC score with the rate of hospital-based mortality and 30-day mortality. Such research would help determine the overall predictive value of the ABC score for the prognosis of patients with UGIB due to esophageal and variceal hemorrhage. Third, it would be valuable to compare the ABC score with other scoring systems such as Rockall and AIMS65 to identify the optimal scoring system or the potential combination of these scoring systems.

## 5. Conclusions

Our study shows the association of the ABC score with the incidence of early rebleeding and in-hospital mortality in cirrhotic patients with variceal bleeding. Further research could be conducted across multiple centers with a larger sample size to enhance the reliability and generalizability of the findings, enabling a more confident application of the ABC score in clinical practice.

## Figures and Tables

**Figure 1 diagnostics-13-03570-f001:**
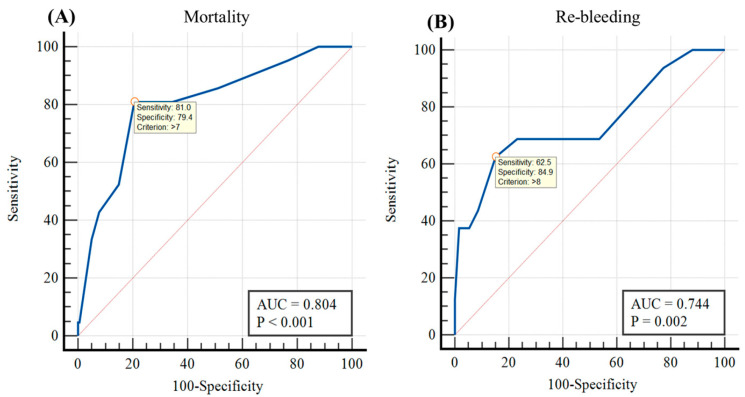
The area under the curve (AUC) of the ABC score in predicting in-hospital mortality (**A**) and early rebleeding (**B**).

**Table 1 diagnostics-13-03570-t001:** Clinical and endoscopic characteristics of the study participants (*n* = 201) at baseline.

Variable		X ± SD or *n* (%)
Age		56.4 ± 10.8
Male		185 (92.0%)
Etiology of cirrhosis	HBV	37 (18.4%)
	HCV	5 (2.5%)
	Alcoholic	115 (57.2%)
	Alcoholic and HBV/HCV	31 (15.4%)
	Others	13 (6.5%)
Clinical symptoms at the time of admission	Hematemesis	47 (23.4%)
	Melena	51 (25.4%)
	Both	103 (51.2%)
Child–Pugh class	A	43 (21.4%)
	B	100 (49.8%)
	C	58 (28.8%)
MELD score	≤15	152 (75.6%)
	>15	49 (24.4)
History of upper gastrointestinal bleeding due to esophageal or gastric variceal rupture	Yes	143 (57.2%)
	No	58 (42.8%)
Esophageal varies	No	4 (2.0%)
	Grade I	4 (2.0%)
	Grade II	31 (15.4%)
	Grade III	162 (80.6%)
Gastric varies	Yes	63 (31.3%)
	No	122 (60.7%)
	Hard observations	16 (8.0%)
Status of bleeding from variceal rupture.	Active	41 (20.4%)
	Inactive	160 (79.6%)

The value of age is given as an average.

**Table 2 diagnostics-13-03570-t002:** The rates of early rebleeding and in-hospital mortality in the study group (*n* = 201).

Events	Number (*n*)	Rate (%)
Early rebleeding	16	8.0
In-hospital mortality	21	10.4

Early rebleeding was defined as recurrent bleeding episodes occurring within the first five days. In-hospital mortality refers to the rate or occurrence of deaths that happen within the hospital during a patient’s admission.

**Table 3 diagnostics-13-03570-t003:** The variables of the ABC score (*n* = 201).

Variables of ABC Score		*n* (%)
Average	6.3 ± 2.5	
Min–Max	3–15	
Risk stratification	Low (*n*. %)	22 (10.9)
	Moderate (*n*. %)	125 (62.2)
	High (*n*. %)	54 (26.9)

The score ranges from 0 to 18 points, categorizing the risk into low (≤3 points), moderate (4–7 points), and high (≥8 points) levels.

**Table 4 diagnostics-13-03570-t004:** The association between risk categories of the ABC score and early rebleeding and in-hospital mortality (*n* = 201).

ABC Scores	Early Rebleeding				In-Hospital Mortality			
	No	Yes	OR (95% CI)	*p* Value	No	Yes	OR (95% CI)	*p* Value
	(*n*. %)	(*n*. %)			(*n*. %)	(*n*. %)		
Average	6.1 ± 2.2	9.1 ± 3.4		<0.001	6.0 ± 2.2	9.1 ± 2.7		<0.001
High risk (*n* = 54)	43 (79.6)	11 (20.4)	1		37 (68.5)	17 (31.5)	1	
Moderate risk (*n* = 125)	120 (96.0)	5 (4.0)	0.8 [0.7–0.9]	0.0004	121 (96.8)	4 (3.2)	0.7 [0.6–0.8]	<0.0001
Low risk (*n* = 22)	22 (100)	0 (0)	0.8 [0.6–0.9]	0.02	22 (100)	0 (0)	0.7 [0.6–0.8]	0.003

The scores ranges from 0 to 18 points, categorizing the risk into low (≤3 points), moderate (4–7 points), and high (≥8 points) levels. Early rebleeding was defined as recurrent bleeding episodes occurring within the first five days. In-hospital mortality refers to the rate or occurrence of deaths that happen within the hospital during a patient’s admission.

**Table 5 diagnostics-13-03570-t005:** The value of the ABC score in predicting early rebleeding and in-hospital mortality.

Events	Cutoff ABC Score	AUROC	95% CI	Sensitivity (%)	Specificity (%)	PPV (%)	NPV (%)	*p* Value
In-hospital mortality	8	0.804	0.70–0.91	81.0	79.4	31.5	97.3	<0.001
Early rebleeding	9	0.744	0.59–0.89	62.5	84.9	26.3	96.3	0.002

In-hospital mortality refers to the rate or occurrence of deaths within the hospital during a patient’s admission. Early rebleeding was defined as recurrent bleeding episodes occurring within the first five days. AUROC, area under the receiver operating characteristics; PPV, positive predictive value; NPV, negative predictive value.

## Data Availability

The datasets generated and/or analyzed during the current study are not publicly available due to the privacy policy of the Vietnam military hospital but are available from the corresponding author upon reasonable request.

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
