# Peer review of "Predictive Significance of the ABC Score for Early Re-Hemorrhage and In-Hospital Mortality in High-Risk Variceal Bleeding among Cirrhotic Patients"

_diagnostics, 2023, doi:10.3390/diagnostics13233570_

Round 1
Reviewer 1 Report
Comments and Suggestions for Authors

Comments on the Quality of English Language
Author Response
|
Comments 1: You should identify the characteristics of patients like etiology of cirrhosis, Child-Pugh group, MELD score, laboratory values etc. in RESULTS section. |
|
Response 1: Thank you for pointing this out. We agree with this comment. We have, accordingly, added some patient characteristics in Table 1, and commented on the result. “[updated text in the manuscript at line 137-140]”
|
|
Comments 2: Please clarify that whether the bleeding episode was related with ESOV/GASV or not. Other causes of GIB (erosions, angiodisplasia, diverticular bleeding etc.) may be responsible from bleeding episode in especially subjects presenting with melena. Please check the endoscopic findings that suggest the bleeding was related to varices like white spot, red spot, clot on varix etc. |
|
Response 2: Agree. We added into the Inclusion criteria at Method section, at line 87-92. “[updated text in the manuscript]”
|
|
Comments 3: The causes of death were not fully disclosed in the study. For example, in the high-risk group,death was observed in 16 people, while re-bleeding was observed in only 11 of them. This suggests that mortality may be due to other causes than bleeding. In this study, it was stated that 20.4% of the patients had active bleeding at the time of admission. Is there any death because of the failure in index endoscopic intervention at admission? |
|
Response 3: Thank you for pointing this out. We agree with this comment. - Not all mortality cases in our patient cohort were due to GI bleed. - Out of 21 patients, 10 died due to severe hemorrhagic shock and profound metabolic disturbances, despite successful hemostatic interventions. In 6 out of 21 patients, there was recurrent bleeding, and the second intervention was unsuccessful. One patient passed away from acute coronary syndrome on the third day after admission. The remaining patients died due to various other reasons, such as infections, hepatic encephalopathy, and organ failure. - We added this information to the Result section, and discussion. “[updated text in the manuscript at line 233-240]”
|
|
Comments 4: Cirrhosis has unique clinical and laboratory features. Therefore, predicting high-risk patients is not as easy as it seems. In ABC scoring, liver cirrhosis gives 2 points. Moreover, most of the cirrhotic patients are evaluated as ASA 3 or ASA 4 class. Hypoalbuminemia is also common among cirrhotic subjects. In the severe forms of cirrhosis, development of HRS or ACLF related with increase in creatinine value. Moreover, abundant GI bleeding from varices is related with excessive BUN increase. All of these factors put cirrhotic patients into the moderate to high-risk groups according to ABC scoring system. Thus, usefulness of ABC score to predict mortality among this special patient group may be doubtful. The authors can elaborate on this topic in the DISCUSSION section |
|
Response 4: Thank you for your comment. We partly agree with it. - Cirrhosis has unique clinical and laboratory features. Therefore, predicting high-risk patients is not as easy as it seems. In the ABC scoring system, liver cirrhosis is assigned 2 points. Furthermore, most cirrhotic patients are evaluated as ASA 3 or ASA 4 class. Hypoalbuminemia is also common among cirrhotic subjects. In the severe forms of cirrhosis, the development of HRS or ACLF is associated with increased creatinine levels. Additionally, substantial gastrointestinal bleeding from varices correlates with a significant rise in blood urea nitrogen (BUN) levels. According to the ABC scoring system, these factors categorize cirrhotic patients into moderate to high-risk groups. Therefore, the utility of the ABC score for predicting mortality among this particular patient group may be subject to doubt. - However, not all cirrhotic patients fit this profile, and it depends on the disease's stage, severity, and progression. Cirrhotic patients may have higher ABC scores than those with gastrointestinal bleeding due to other causes. Nevertheless, the ABC score retains its value in predicting early re-bleeding and mortality in this patient group, with a corresponding prediction cutoff point for each outcome. In our study, the ABC score demonstrated good predictive value for early mortality with an area under the receiver operating characteristic curve (AUROC) of 0.804; 95% CI: 0.70-0.91; p<0.001; at a cutoff point of 8, it showed a sensitivity of 81% and specificity of 79.4%. It had a reasonably good predictive value for early re-bleeding with an AUROC of 0.744; 95% CI: 0.59-0.89, p=0.002. At a cut-off point of 9, it had a sensitivity of 62.5% and a specificity of 84.9%. - We have added this discussion to the section on line 260-273 “[updated text, and reference in the manuscript at line ]” |
|
Comments 5: Table 1. Gastric varices – Challenging 16 (8%). What did you mean as'Challenging''? |
|
Response 5: Thank you for your comment. - “Challenging” means hard observations - I have, accordingly, modified the word in the table 1 “[updated text, and reference in the manuscript at line ]” |
|
Comments 6: Line 280: ''the preliminary results of this study demonstrate …'' should be corrected in CONCLUSIONS section |
|
Response 6: Thank you for your comment. We have, accordingly, modified this point in the conclusion section, at line 314-318 “[updated text, and reference in the manuscript ]” |
|
4. Response to Comments on the Quality of English Language |
|
Point 1: Would be beneficial to use medical terminology rather than common terms |
|
Response 1: Agree. These words were changed: hematemesis, melena, in-hospital mortality |
|
5. Additional clarifications |
|
[Here, mention any other clarifications you would like to provide to the journal editor/reviewer.] |

Reviewer 2 Report
Comments and Suggestions for Authors
The paper is presenting an observational study which verifies the previously published ABC bleeding score. The authors find that the ABC score is accurate in predicting in-hospital death and acceptable in predicting early re-bleeding.
It is in my opinion adequately covering the subject using a small case series from the own medical center to confirm data published from other geographic areas. The paper is only referring to the ABC score, it lacks comparison with other predictive scoring systems.
The structure of the manuscript could in my opinion be improved. Some information (e.g., demographic data) is provided more than once in the same form.
The cited references are relevant to the topic. The figures are clear and easy to understand, although some of the tables contain spelling errors that need to be corrected (e.g., Table 1).
I find the conclusions are consistent with the argumentation; they are in concordance with previous published data. The conclusions are supported by the listed citations.
The work is no novelty on the field but confirmatory, it could raise awareness among the physicians that patients with variceal bleeding can be better selected with regards of the risk for early complications and presumably better monitored to improve the outcome. There are, to my knowledge, only a few papers covering the value of the ABC score in patients with liver cirrhosis and variceal bleeding.
I suggest a minor revision of the language, also a review of the whole manuscript as it still contains standardized sentences from the template.
Comments on the Quality of English Language
Some of the tables contain spelling errors that need to be corrected (e.g., Table 1).
I suggest a minor revision of the language, also a review of the whole manuscript as it still contains standardized sentences from the template.
Author Response
Thank you for your comments,
I already corrected spelling errors and rewrote some sentences. A new version looks better. Please find it in the attached file.
Reviewer 3 Report
Comments and Suggestions for Authors
More recent references should be added.
Comments on the Quality of English Language
Minor English language corrections are still needed
Author Response
|
1. Point-by-point response to Comments and Suggestions for Authors |
|
Comments 1: More recent references should be added. |
|
Response 1: Thank you for pointing this out. I/We agree with this comment. Therefore, I/we have to add more recent references. “[updated text in the manuscript if necessary]”
|
|
2. Response to Comments on the Quality of English Language |
|
Point 1: Minor English language corrections are still needed |
|
Response 1: Agree. We have, accordingly, rewrote some sentences to emphasize this point. |
|
3. Additional clarifications |
|
[Here, mention any other clarifications you would like to provide to the journal editor/reviewer.] |

Reviewer 4 Report
Comments and Suggestions for Authors
Authors in this study have used ABC score in their cohort and analyzed its efficacy.
Methods: Please mention that this is a retrospective study.
Please mention Ethics or IRB statement.
Rebleeding: what is the upper limit in terms of duration - 5 days? is it same for definition and in the results analyzed.?
Why a drop of 3g/dl was used? what is the rationale? Any change in vitals taken into account?
What about patients with transfusion and drop in Hb?
Please use Hematemesis or melena rather than vomiting blood or black colored stools. In hospital mortality would be better term for internal hospital death.
What is medical central death?
How was hemostasis achieved? banding vs sclerotherapy vs hemospray vs coiling among others? Rebleeding depends on these interventions as well. ? How many had coagulopathy? how many had liver dysfunction? any renal disorders.
Any other clotting factors deficiency? Anyone on anticoagulant?
The original score is for mortality prediction, without external validation this cannot be extrapolated for rebleeding.
Discussion: Please address issues in methods and reshape accordingly.,
Comments on the Quality of English Language
Would be beneficial to use medical terminology rather than common terms.
Author Response
|
Comments 1: Please mention that this is a retrospective study. |
||
|
Response 1: [Type your response here and mark your revisions in red] Thank you for pointing this out. I/We agree with this comment. The study design was a retrospective study. Therefore, I/we have added this information to the manuscript “[updated text in the manuscript at line 98]”
|
||
|
Comments 2: Please mention Ethics or IRB statement |
||
|
Response 2: Agree. We have accordingly added more details in part of the Institutional Review Board Statement, at line 330-331. “[updated text in the manuscript at line 330-331]” |
||
|
Comments 2: Rebleeding: what is the upper limit in terms of duration - 5 days? is it same for definition and in the results analyzed.? |
||
|
Response 2: Thank you for pointing this out. I/We partly agree with this comment. - Rebleeding was defined as the occurrence of hematemesis after 2 hours or more post-treatment with medication or endoscopic hemostasis, or a decrease in Hemoglobin levels by 30g/L or more (approximately 9% Hematocrit) within 24 hours in patients who did not receive blood transfusion (reference 11) (line 109-113 in the manuscript). - Early rebleeding was defined occurrence of hematemesis episodes within the first five days (line 109-113 in the manuscript) - We chose the duration of the first 5 days from administration as a time point of early rebleeding, according to the consensus of Baveno VII, at point 6.14. It said that rebleeding within the first 5 days is defined as a five-day treatment failure; so these patients might be threatened with life. - These definitions were used to analyze all data. “[updated text in the manuscript at line 109 – 113, and updated references]”
|
||
|
Comments 3: Why a drop of 3g/dl was used? what is the rationale? Any change in vitals taken into account? What about patients with transfusion and drop in Hb? |
||
|
Response 3: Thank you for pointing this out. - A drop of 3g/dl was used following the consensus of Baveno V, which identified the rebleeding criteria. This rationale was also used to analyze the prospective validation of Baveno V definitions and criteria for failure to control bleeding in Portal Hypertension (Ahn, S.Y., et al., Prospective validation of Baveno V definitions and criteria for failure to control bleeding in portal hypertension. Hepatology, 2015. 61(3): p. 1033-40). I updated the references for this point. - Patients with transfusion and drop in Hb: The patients were transfused until their hemoglobin ≥ 8g/dl. After that, we provide internal medicine treatment and monitor hemoglobin levels every 6-8 hours. If hemoglobin drops below 6g/dl, we assess it as recurrent bleeding and administer a blood transfusion. The patient would undergo endoscopy again to evaluate recurrent bleeding if the hemoglobin level decreases rapidly. “[updated text, and reference in the manuscript at line 109-114]”
|
||
|
Comments 4: Please use Hematemesis or melena rather than vomiting blood or black colored stools. In hospital mortality would be better term for internal hospital death. What is medical central death? |
||
|
Response 4: Agree. We have, accordingly, changed words to emphasize this point. - “Medical central death” means in-hospital mortality “[updated text in the manuscript]”
|
||
|
Comments 5: How was hemostasis achieved? banding vs sclerotherapy vs hemospray vs coiling among others? Rebleeding depends on these interventions as well. ? How many had coagulopathy? how many had liver dysfunction? any renal disorders. |
||
|
Response 5: Agree. I/We have, accordingly, done/revised/changed/modified…..to emphasize this point. - All patients in the study population underwent endoscopy and hemostatic intervention following the national guidelines: Esophageal variceal bleeding was managed by rubber band ligation, and esophageal variceal bleeding was treated with injection of varices using Histoacryl. (This point was updated to the manuscript) - All patients were treated using a standardized bleeding control method, so the recurrence of bleeding depended on the individual patient's prognostic factors. - Among 201 study patients, 176 had a history of liver cirrhosis (Table 1), but only 13 patients had an INR > 2.3 - There were 17 patients with liver dysfunction (INR > 1.5 and encephalopathy) - There were 31 patients with renal disorders. However, the creatinine level and liver cirrhosis are part of the ABC score used for calculation and analysis. I added more discussion at line 260-273 related to this point “[updated text in the manuscript at line 98-101]”
|
||
|
||
|
Comments 7: The original score is for mortality prediction, without external validation this cannot be extrapolated for rebleeding. |
||
|
Response 7: Thank you for your input. We completely agree with this perspective. Therefore, in the study design, we excluded patients with confounding factors that could not be determined to affect the outcome analysis. We also added some relevant items in the method section. “[updated text in the manuscript if necessary]”
|
||
|
Comments 8: Discussion: Please address issues in methods and reshape accordingly |
||
|
Response 8: Thank you for pointing this out. I/We agree with this comment. Therefore, I/we have modified the method section to emphasize this point. ““[updated text in the manuscript if necessary]” |
||
|
4. Response to Comments on the Quality of English Language |
||
|
Point 1: Would be beneficial to use medical terminology rather than common terms |
||
|
Response 1: Agree. These words were changed: hematemesis, melena, in-hospital mortality |
||
|
5. Additional clarifications |
||
|
[Here, mention any other clarifications you would like to provide to the journal editor/reviewer.] |
Round 2
Reviewer 1 Report
Comments and Suggestions for Authors
I thank to Editors for reviewing this study. This study reveals the prognostic role of the ABC score in predicting mortality and re-bleeding in cirrhosis patients. There are some concerns about the revised version of the manuscripts.
Comments:
1- In Results section and Table 1, sum of the patients in Child groups is higher than 201.
2- There is also a discrepancy in the total number of patients according to the causes of cirrhosis in Table 1.
3- According to the results of this study, more than half of the patients in Child group A. Moreover, alcoholic hepatitis seems the cause of cirrhosis in more than half of the patients. Alcoholic hepatitis resembles cirrhosis in terms of clinical features like ascites, splenomegaly and varices and, laboratory findings including hypoalbuminemia, anemia, thrombocytopenia etc. So, it is not always easy to distinguish between these two entity and, sometimes histopathological assessment is required for definitive diagnosis. How can the authors exclude alcoholic hepatitis, especially in Child A group?
4- Line 251: You may use ‘‘in patients’‘ instead of ‘‘patients’’.
5- You may use ‘‘upper’‘ instead of ‘‘high’’ in Discussion section (Lines 253 and 267).
Comments on the Quality of English Language
The English is well-written.
Author Response
|
Comments 1: In Results section and Table 1, sum of the patients in Child groups is higher than 201. |
|
Response 1: Thank you for pointing this out. We agree with this comment. We have, accordingly, corrected some points in Table 1, and re-written the result due to my typing errors. “[updated text in the manuscript at line 138]”
|
|
Comments 2: There is also a discrepancy in the total number of patients according to the causes of cirrhosis in Table 1 |
|
Response 2: Agree. We apologize for this mistake. We edited several points in Table 1. “[updated text in the manuscript]”
|
|
Comments 3: According to the results of this study, more than half of the patients in Child group A. Moreover, alcoholic hepatitis seems the cause of cirrhosis in more than half of the patients. Alcoholic hepatitis resembles cirrhosis in terms of clinical features like ascites, splenomegaly and varices and, laboratory findings including hypoalbuminemia, anemia, thrombocytopenia etc. So, it is not always easy to distinguish between these two entity and, sometimes histopathological assessment is required for definitive diagnosis. How can the authors exclude alcoholic hepatitis, especially in Child A group? |
|
Response 3: Thank you for pointing this out. We partly agree with this comment. - Patients participating in the study were diagnosed with liver cirrhosis based on the national guidelines. These patients had a history of chronic liver damage (for example: long-term alcohol use, and chronic hepatitis), and presented clinically with both impaired liver function and portal hypertension syndrome. In case of necessity, Fibroscan was used to evaluate stages of liver fibrosis, and only patients with fibrosis score F3- F4 were included. A liver biopsy is not mandatory for diagnosing liver cirrhosis in this clinical setting. - We added this information to the Method section. “[updated text in the manuscript at line 87-91]”
|
|
Comments 4: Line 251: You may use ‘‘in patients’‘ instead of ‘‘patients’’. |
|
Response 4: Thank you for your comment. We partly agree with it. - We did not find this point as your comment from line 249 to line 262. “[updated text, and reference in the manuscript at line ]” |
|
Comments 5: You may use ‘‘upper’‘ instead of ‘‘high’’ in Discussion section (Lines 253 and 267). |
|
Response 5: Thank you for your comment. - We have corrected this word to the discussion section on line 258 and line 295 “[updated text, and reference in the manuscript at line ]” |

Reviewer 4 Report
Comments and Suggestions for Authors
Authors have answered my queries and I recommend to accept the paper. Thanks
Comments on the Quality of English Language
Minor spell checks.
Author Response
Thank you in advance for your comments and suggestions.
Round 3
Reviewer 1 Report
Comments and Suggestions for Authors
Dear Authors,
I thank to Editors for reviewing this study. It was seen that the required changes and corrections were made by the authors in the revised manuscript.
Sincerely